# Distributed Optical Fiber-Based Approach for Soil–Structure Interaction

**DOI:** 10.3390/s20010321

**Published:** 2020-01-06

**Authors:** Nissrine Boujia, Franziska Schmidt, Christophe Chevalier, Dominique Siegert, Damien Pham Van Bang

**Affiliations:** 1Université Paris Est, Ifsttar, 77447 Champs sur Marne, France; nissrine.boujia@ifsttar.fr (N.B.); christophe.chevalier@ifsttar.fr (C.C.); dominique.siegert@ifsttar.fr (D.S.); 2INRS, Centre Eau Terre Environnement, Québec, QC G1K 9A9, Canada; damien.pham_van_bang@ete.inrs.ca

**Keywords:** distributed measurements, fiber optic sensors, scour, soil-structure interaction, winkler model, equivalent length

## Abstract

Scour is a hydraulic risk threatening the stability of bridges in fluvial and coastal areas. Therefore, developing permanent and real-time monitoring techniques is crucial. Recent advances in strain measurements using fiber optic sensors allow new opportunities for scour monitoring. In this study, the innovative optical frequency domain reflectometry (OFDR) was used to evaluate the effect of scour by performing distributed strain measurements along a rod under static lateral loads. An analytical analysis based on the Winkler model of the soil was carefully established and used to evaluate the accuracy of the fiber optic sensors and helped interpret the measurements results. Dynamic tests were also performed and results from static and dynamic tests were compared using an equivalent cantilever model.

## 1. Introduction

Bridge scour occurs when flowing water erodes sediments around bridge supports, more precisely piers and abutments. When scour depth reaches a critical value, the stability of the bridge is threatened which may lead to its collapse. Therefore, it is crucial to develop monitoring techniques capable of assessing scour depth in order to anticipate this hydraulic risk. Over the last twenty years, scour monitoring technologies have evolved significantly. Beginning with the use of traditional geophysical instruments such as radar [1] and sonar [2] to developing new sensors [3] and the use of more sophisticated tools such as fiber Bragg grating sensors [4], scour monitoring is an ongoing topic of research.

Over the past few years, the effect of scour on the static response of single piles has gained interest and has been reported by several studies: Lin et al. [5] studied the effect of scour on the response of laterally loaded piles considering the change of stress of the remaining sand. Qi et al. [6] investigated the effect of local and global scour on the p-y curves of piles in sand using various centrifuge model tests. These studies showed that scour induces changes in the response of laterally loaded piles. However, the use of those changes to determine scour depth is still limited. An example of those uses is provided in [7]: A scour sensor was equipped with fiber Bragg grating (FBG) sensors. The monitoring system consists of a cantilever beam equipped with FBG sensors having different wavelengths and placed at various heights. The sensor operates on the principle that as long as a FBG sensor is embedded in the soil, the registered value of strain at the sensor location is negligible. Once scour occurs, the FBG sensor emerges and is subjected to water flow. Consequently, the measured value of strain increases. Thus, knowing the height of the sensors having registered the variation of strain, scour depth can be determined. A major limitation of this scour sensor is that FBG sensors can only perform quasi-distributed monitoring and their number along the fiber is limited [8].

The dynamic monitoring of scour has also become increasingly important in recent years. Many authors suggest monitoring the modal parameters of the structure itself (i.e., spans or piers) [9,10,11]. Other authors suggest monitoring the vibration frequency of sensors rods embedded in the riverbed [4,12]. A numerical model, based on the Winkler theory of the soil, is then usually used to establish a relationship between the measured frequency and scour depth. One of the main challenges of the dynamic monitoring technique is the difficulty in evaluating the modulus of subgrade reaction $K_{s}$ through a rational and methodical approach. The value of $K_{s}$ depends not only on the Young modulus of the soil $E_{s}$, but also on various geometric and mechanical parameters of the structure itself.

In this study, the effect of scour on both the static and dynamic responses of a rod/sensor is studied. The optical frequency domain reflectometry (OFDR) technique is used to measure strain to overcome the limitations of FBG sensors. OFDR technology enables measuring strain along structures with millimeter level spacial resolution. A rectangular rod was instrumented along its length with a distributed fiber optic strain sensors (OFDR). The rod-sensor was then tested under static lateral loads for different scour depths. These tests were conducted under tension and compression to make sure that the results are independent of the testing configuration. An analytical model, based on the Winkler soil model was then developed to help in static results interpretations. The key parameter of the proposed model is the modulus of subgrade reaction $K_{s}$. In this study, its value was determined from Ménard tests and was further confirmed with the fiber optic measurements. Dynamic tests were also conducted in the same testing conditions. Finally, an equivalent cantilever model is proposed in order to compare the static and dynamic approaches used in this study.

The paper starts with a description of the experimental protocols for static and dynamic tests. A second part highlights the main results of this study and presents the analytical model for the static experiments. A third part introduces introduces a simplified cantilever based model which allows modeling the soil–structure interaction for static and dynamic experiments.

## 2. Theoretical Formulation

The Winkler approach [13,14,15] was used to model the static experimental tests. According to this approach, it is assumed that the beam is supported by a series of infinitely closed independent and elastic springs. The governing equation of a laterally loaded beam, partially embedded in the soil, is expressed by Equation (Equation 1):(1)$$\left\{\begin{array}{cc} E_{b}I_{b}\frac{d^{4}w\left(z\right)}{d^{4}z}=0 & \mathrm{for}\;z\in [-a,0],\\ E_{b}I_{b}\frac{d^{4}w\left(z\right)}{d^{4}z}+K_{s}w\left(z\right)=0 & \mathrm{for}\;z\in [0,D], \end{array}\right.$$
where $E_{b}$ and $I_{b}$ are the Young modulus and the cross section moment of inertia of the beam respectively, w(z) the lateral deflection of the beam, $K_{s}$ the modulus of subgrade reaction of the soil, *a* the eccentricity of the load *F* and *D* the embedded length of the beam (see Figure 1).

If $K_{s}$ is constant along the depth, the general solution of this set of equations is given by:(2)$$\left\{\begin{array}{ccc} w_{1}\left(z\right) & =a_{1}z^{3}+a_{2}z^{2}+a_{3}z+a_{4} & \mathrm{for}\;z\in [-a,0],\\ w_{2}\left(z\right) & =e^{\left(-z/l_{0}\right)}\left(a_{5}cos\left(z/l_{0}\right)+a_{6}sin\left(z/l_{0}\right)\right) & \mathrm{for}\;z\in \left[0,D\right]\\ \; & +e^{\left(z/l_{0}\right)}\left(a_{7}cos\left(z/l_{0}\right)+a_{8}sin\left(z/l_{0}\right)\right), \end{array}\right.$$
where the characteristic length of the beam is $l_{0}={\left(\frac{4E_{b}I_{b}}{K_{s}}\right)}^{\frac{1}{4}}$. It is worth noting that $l_{0}$ combines mechanical properties of both the soil and the pile.

For the case of flexible or long piles, as the rod used in this study, positive exponential terms in Equation (Equation 2) are negligible. Consequently $a_{7}=a_{8}=0$ and only six parameters remain to be determined. The following notations are used to refer to the bending moment $M_{i}=E_{b}I_{b}w_{i}^{\prime \prime }$, the shear force $T_{i}=E_{b}I_{b}w_{i}^{\prime \prime \prime }$ and the slope $\theta_{i}=w_{i}^{\prime }$.

By assuming that the displacements are small and that the applied force is strictly perpendicaular to the beam, boundary conditions on moment and shear force for z=−a can be written: $M\left(-a\right)=\;E_{b}I_{b}{w_{1}}^{\prime \prime }\left(-a\right)=0$ and $T\left(-a\right)=E_{b}I_{b}{w_{1}}^{\prime \prime \prime }\left(-a\right)=F$ which gives simple relation of $a_{1}$ and $a_{2}$: $a_{1}=\frac{F}{6E_{b}I_{b}}$ and $a_{2}=\frac{-Fa}{2E_{b}I_{b}}$. Finally, adding four equations expressing the continuity of displacement *w*, slope θ, bending moment *M* and shear force *T* at the ground surface z=0 gives enough constraints:(3)$$\left\{\begin{array}{c} w_{1}\left(0\right)=w_{2}\left(0\right)\\ \theta_{1}\left(0\right)=\theta_{2}\left(0\right)\\ M_{1}\left(0\right)=M_{2}\left(0\right)\\ T_{1}\left(0\right)=T_{2}\left(0\right). \end{array}\right.$$

These conditions can be written as
[M]{X}={A} where:
$$A=\left(\begin{array}{c} 0\\ 0\\ \frac{Fa}{E_{b}I_{b}}\\ \frac{F}{E_{b}I_{b}} \end{array}\right),X=\left(\begin{array}{c} a_{3}\\ a_{4}\\ a_{5}\\ a_{6} \end{array}\right)$$
$$M=\left(\begin{array}{cccc} 0 & -1 & 1 & 0\\ -1 & 0 & -\frac{1}{l_{0}} & \frac{1}{l_{0}}\\ 0 & 0 & 0 & -\frac{2}{{l_{0}}^{2}}\\ 0 & 0 & \frac{2}{{l_{0}}^{3}} & \frac{2}{{l_{0}}^{3}} \end{array}\right).$$

The solution of the system is:(4)$$a_{6}=-\frac{F{l_{0}}^{2}}{2E_{b}I_{b}}a,a_{3}=\frac{F(2a+l_{0})}{2E_{b}I_{b}}l_{0},a_{4}=a_{5}=\frac{F{l_{0}}^{2}}{2E_{b}I_{b}}\left(a+l_{0}\right).$$

The value of the bending moment along the rod is therefore given by Equation (Equation 5):(5)$$\left\{\begin{array}{cc} M_{1}\left(z\right)=E_{b}I_{b}{w_{1}}^{\prime \prime }\left(z\right)=E_{b}I_{b}\left(6a_{1}z+2a_{2}\right)=F\left(z+a\right) & z\in [-a,0],\\ M_{2}\left(z\right)=E_{b}I_{b}{w_{2}}^{\prime \prime }\left(z\right)=F\exp\left(\frac{-z}{l_{0}}\right)\left(l_{0}\sin\left(\frac{z}{l_{0}}\right)+a(\cos\left(\frac{z}{l_{0}}\right)+\sin\left(\frac{z}{l_{0}}\right)\right) & z\in \left[0,D\right]. \end{array}\right.$$

The normal stress induced by the bending moment in the rod at the outermost fibers is given by the known formulation:(6)$$\sigma =\pm \frac{M}{I_{b}}\times \frac{h}{2},$$
where $\frac{h}{2}$ is the distance from the neutral axis to the outermost fibers. The linear elastic strain along the rod is then deduced using the Hooke’s law:(7)$$\epsilon =\pm \frac{M}{E_{b}I_{b}}\times \frac{h}{2}.$$

Moreover, the expression of the strain along the beam is given by the following equation:(8)$$\begin{array}{ccc} \epsilon & = & \frac{M}{E_{b}I_{b}}z,\\ \; & = & \frac{E_{b}I_{b}\left(6a_{1}z+2a_{2}\right)}{E_{b}I_{b}}z=6a_{1}z+2a_{2},\;\mathrm{for}\;z\in \left[-a,0\right]\\ \; & = & \frac{F\exp\left(\frac{-z}{l_{0}}\right)\left(l_{0}\sin\left(\frac{z}{l_{0}}\right)+a(\cos\left(\frac{z}{l_{0}}\right)+\sin\left(\frac{z}{l_{0}}\right)\right)}{E_{b}I_{b}}z,\;\mathrm{for}\;z\in \left[0,D\right]. \end{array}$$

The positions of extreme strain values verify $M^{\prime }\left(z\right)=0$. Therefore, the position of the maximum strain in the embedded part of the rod $z_{max}$ is given with Equation (Equation 9) and varies with the eccentricity of the load *a* and the characteristic length $l_{0}$:(9)$$z_{max}=l_{0}\arctan\left(\frac{l_{0}}{l_{0}+2a}\right).$$

## 3. Fiber Optic Sensing Technology

The sensing technology used to measure strains along the rod is optical frequency domain reflectometry. OFDR enables measurement along a fiber up to 2 km long, with millimetre level spacial resolution [16]. The light emitted from a highly tunable laser source undergoes a coupler and is then divided between two branches: the reference branch and the fiber under test branch. Backscattered lights from both branches are then combined to create an interference signal. This signal is detected by an optical detector. The Rayleigh backscattering induced by the random fluctuations in the refractive index along the fiber length can be modelled as a Bragg grating with random period [17]. As long as the fiber is in a stable state, the Rayleigh backscattering spectrum remains constant. When the surrounding environment of the fiber changes due to external stimulus (as strain and temperature), a spectrum shift occurs. This spectrum shift is expressed using Equation (Equation 10):(10)$$\Delta \nu =C_{\epsilon }\times \epsilon +C_{T}\times \Delta T,$$
where Δν is the Rayleigh spectral shift, ϵ the fiber strain, ΔT the temperature variation of the fiber, $C_{\epsilon }$ and $C_{T}$ are calibration constants. The typical values of the latter parameters for a standard single-mode fiber, at 1550 nm, are respectively: −0.15 GHz/μϵ and −1.25 GHz/C°. This strain/temperature dependent spectrum shift can be determined by means of cross correlation between reference scan (meaning the scan performed at ambient temperature and null strain state) and measurement scan (when a temperature perturbation or a strain is applied). Fundamental principles of Rayleigh systems are fully detailed in [18].

In the experimental testing, a fiber optic sensor was glued along the length of the rod (Figure 1). A two-component Methyl Methacrylate paste was used as an adhesive. The commercially available optoelectronic optical backscatter reflectometer (OBR) from Luna Technology [19] was used. Th spatial resolution used during the experiment is 5 mm leading to detailed strain profiles along the rod. The acquisition time of the fiber optic sensing signal is 5 s, which limits its use to static testing.

### Experimental Setup

The experimental setup used to perform the static and dynamic tests on the rod is shown in Figure 1. A rigid tank of dimensions 1 m × 1 m × 1 m was progressively filled with dry sand reaching a final height of 0.7 m. The static lateral loads were applied by a set of dead weights connected to the rod described in Section 4.1.1 with a thread passing through a pulley. The static lateral loads were applied with various eccentricities *a*. To control the direction, and therefore the value of the lateral loads, the height of the pulley was adjustable in order to keep the part of the thread connected to the rod always in horizontal position.

## 4. Materials and Methods

### 4.1. Materials Used

#### 4.1.1. Rod Characteristics

An aluminum rectangular rod having width b=25 mm, thickness h=5 mm and length L=1170 mm was used. Its bulk density and Young modulus were respectively $\rho_{b}=2700$ kg/m^3^ and $E_{b}=62.2$ GPa. The rod was instrumented using a fiber optic along its length to measure the strain. Following [20], an accelerometer was placed at the head of the rod, to record its transient response.

#### 4.1.2. Soil Characteristics

The static and dynamic tests were conducted using a dry sand of Seine. The mean size of sand grains was $D_{50}=0.70$ mm and the dry density was $\rho_{s}=1700$ kg/m^3^. For soil characterization, mini-pressuremeter tests were performed. The average Ménard modulus measured in these tests was $E_{m}=0.5$ MPa. The subgrade modulus $K_{s}$ of the tested soil could then be calculated using the empirical Equation (Equation 11) [21]:(11)$$\left\{\begin{array}{cc} K_{s}=\frac{3E_{m}}{\frac{2}{3}\left(\frac{B}{B_{0}}\right)\left(\frac{\alpha }{2}\right){\left(2.65\frac{B}{B_{0}}\right)}^{\alpha }} & B>B_{0},\\ K_{s}=\frac{18E_{m}}{4{\left(2.65\right)}^{\alpha }+3\alpha } & B<B_{0}, \end{array}\right.$$
where $K_{s}$ is the modulus of subgrade reaction, $E_{m}$ is the Ménard modulus, *B* is the diameter of the tested pier or rod, $B_{0}=0.6$ m is the reference diameter and α is a rheological parameter depending on the tested soil with $\alpha =\frac{1}{3}$ for sand. Under these assumptions of parameter values, and given the geometry of the rod, the measured modulus of subgrade reaction of the tested soil is given by Equation (Equation 12):(12)$$E_{s}=1.3\;MPa.$$

### 4.2. Test Procedures

#### 4.2.1. Static Tests

The rod was tested under static lateral loads. The following section provides a description of the testing protocol.

Before applying the lateral loads, a reference scan of the fiber optic was performed. A lateral load *F* was then applied by adding a set of dead weights. A second scan was performed to measure the resulting strain along the rod. Scour was generated by the excavation of a 100 mm thick layer of soil. The various tested configurations are summarized in Table 1. Two loads $F_{1}=2$ N and $F_{2}=4$ N were applied. It is worthy to mention that even if the applied loads were low, the range of generated strains is similar to the usual values of strains along piles.

The tests presented in Table 1 were first performed with the fiber optic under tension, as it was glued on the side of the rod undergoing experiencing a positive bending moment. But the performance of strain sensors may vary due to the testing configuration, e.g., when tested under tension or under compression [22]. The rod was therefore flipped and similar tests were carried out with the fiber optic under compression to evaluate its performance in both configurations.

#### 4.2.2. Dynamic Tests

Free vibration tests were conducted to measure the frequency of the rod for various scour depth. The following section provides a description of the testing protocol.

The rod was partially embedded in sand. Scour was generated by the progressive excavation of 50 mm thick layers of soil. The embedded length *D* of the rod varied from 400 mm to 150 mm. For each scour depth, the thread connecting the rod head to the weight was cut inducing the vibration of the rod in the X direction (horizontal). The signal recorded by the accelerometer was then post-processed using a fast Fourier transform (FFT) to measure the first frequency for each scour depth.

## 5. Results

### 5.1. Static Testing

#### 5.1.1. Fiber Optic Sensor Performance

The results of the lateral loading tests, for the various configurations presented in Table 1, are shown in Figure 2 and Figure 3. The strains obtained under tension are referred to with ‘T’ and the ones obtained under compression are referred to with ‘C’. Figure 2a,b shows the results of tests conducted with the force *F* applied at a distance c=5 cm from the tip of the rod. Figure 3a,b shows the tests for a distance c=25 cm from the tip of the rod.

It is found that the strain curves obtained with the fiber under compression and tension are similar, proving that the fiber optic performance is not affected by its configuration here.

The strain profile along the exposed part of the rod (i.e., for z∈[−a,0]) was proven to be independent of the soil properties. Therefore, the theoretical strain profile is used to evaluate the accuracy of the fiber optic measurement. Theoretical strain profiles for z∈[−a,0] are computed using Equations (Equation 5) and (Equation 7), and compared to the strain measurements with the fiber optic.

As observed in Figure 2 and Figure 3, the experimental and theoretical results are in good agreement. It can be noted that the measurement error does not exceed 7% which highlights the accuracy of the fiber optic sensing technology.

#### 5.1.2. Effect of Scour

As shown in Figure 2 and Figure 3, the strain profiles have a turning point near the ground level plotted with a dashed line. A similar observation was made by [23] who suggested monitoring the maximum bending moment to estimate scour depth.

The static test results also indicate that as scour increases, the bending moment and strain values increase as a consequence of a greater eccentricity of the applied force *F* (due to the higher exposed length of the beam). However, it can be noted that the effect of scour on the strain profiles is only noticeable near the ground level. As the depth increases, no variation of strain values with scour is noticed.

#### 5.1.3. Experimental Strain Profile Versus Theoretical Prediction

The soil layer used during the experiments was modeled using the Winkler model presented previously. The strain along the rod was then computed using Equations (Equation 5) and (Equation 7). The value of the modulus of subgrade reaction $K_{s}$ was determined from mini pressuremeter tests and its value was previously established in Equation (Equation 12).

Figure 2 and Figure 3 show the comparison between the measured and the theoretical strains along the rod. The results show a very good agreement between the theoretical strain profile and the experimental results. These results confirm, on the one hand that its is legitimate to model the soil used in this study as a single layer having a constant subgrade modulus $K_{s}$ with depth, and on the other hand the measured value of the subgrade modulus was also validated as a good agreement was found between the theoretical and experimental strains.

The theoretical positions of the maximum strain are computed using Equation (Equation 9) and summarized in Table 2. As it can be seen, the theoretical results confirm that the maximum bending moment is near the ground level for all testing configurations and gives an insight of the parameters that influence its location which are: the eccentricity of the lateral load *a* and the characteristic length of the rod $l_{0}$.

For a given eccentricity *a* of the lateral load, the variation of $z_{max}$ position with the characteristic length $l_{0}$ can be evaluated by deriving Equation (Equation 9):(13)$$\frac{\partial z_{max}}{\partial l_{0}}=\frac{2al_{0}}{{\left(l_{0}+2a\right)}^{2}+{l_{0}}^{2}}+\arctan\left(\frac{l_{0}}{l_{0}+2a}\right)\geq 0.$$

Therefore, as shown by Equation (Equation 13), the value of $z_{max}$ increases with the increase of the characteristic length $l_{0}$. For this reason, monitoring scour depth using the position of the maximum bending moment/strain can not be generalized for all types of soil and rods. To successfully implement this monitoring technique, it is therefore crucial to carefully design the sensor. The material and geometry in particular should be chosen according to the soil stiffness in order to decrease the characteristic length $l_{0}$ and therefore the value of $z_{max}$.

## 6. Discussion

In this section, the equivalent cantilever model is introduced using a static approach.

### 6.1. Static Equivalent Length

This cantilever has a length $L_{es}$ and a similar deflection to that of the rod partially embedded in sand. The equivalent length $L_{es}$ therefore corresponds to the free length of the rod in sand *H*, increased with a “adjustment static length” *d* corresponding to the distance between the soil level and the equivalent cantilever base as shown in Figure 4.

The methodology of identifying the “adjustment static length” *d* is detailed hereafter.

First, for each tested configuration, the theoretical model is used to determine the deflection along the rod in the sand. To this end, Equation (Equation 2) is used to compute the deflection w(z=−a) at the point of force application.

Second, $L^{\prime }$ is calculated using Equation (Equation 14) derived from the Euler-Bernoulli beam theory.
(14)$$L^{\prime }=\sqrt[{3}]{\frac{3E_{b}I_{b}}{F}\times w\left(-a\right)}.$$

Finally, the position of the base $z_{e}$ of the equivalent cantilever (Figure 4) corresponding to the “adjustment static length” can then be determined using the following equation:(15)$$z_{e}=d=L^{\prime }-a.$$

The previous methodology was applied to the tested configurations and the results are summarized in Table 3. The results show that for all tested configurations, the “adjustment static length” is $z_{e}=d=8.4$ cm. Therefore, the rod in the sand is equivalent to a cantilever beam having a total length $L_{es}=H+d\;(d=8.4$ cm), where *H* the exposed length of the rod and the *d* the “adjustment static length”. This result means that for a given range a embedded length, the partially embedded beam can be considered statically as a cantilevered beam of a higher length, to take into account the embedded length that is needed to support the beam.

It is worthy to highlight that in the case of a fixed cantilever, the point $z=z_{e}=d$ will also have the maximum bending moment which is not the case for the rod partially embedded in sand. Previous results showed that the maximum bending moment is at the ground level.

### 6.2. Dynamic Testing of the Effect of Scour

The variation of the first frequency with the embedded length *D* of the rod is shown is Figure 5. As scour increases, the embedded length *D* of the rod decreases leading to a decrease of the first frequency of the rod. A 10 cm scour from D=40 cm to D=30 cm caused a 20% variation of the frequency.

### 6.3. Dynamic Equivalent Length

The variation of the first frequency with the exposed length *H* of the rod was compared to the the frequencies computed using Equation (Equation 16) of a cantilever carrying a tip mass *m* modelling the accelerometer [24,25].
(16)$$f=\frac{1}{2\pi }\times \sqrt{\frac{3E_{b}I_{b}}{{L_{ed}}^{3}\left(0.24M+m\right)}},$$
where *m* the mass of the accelerometer and *M* the total mass on the cantilever.

Figure 6 shows that the experimental frequencies are translated against the analytical frequencies of an equivalent cantilever with a free length $L_{ed}=H+d^{\prime }$ where $L_{ed}$ is the length of the equivalent cantilever, *H* the exposed length of the rod and $d^{\prime }$ the “adjustment dynamic length” [26]. The latter, $H_{c}$, is determined graphically by shifting the experimental frequency curve to fit the theoretical one (the dotted curve in Figure 6). A good agreement with the theoretical frequencies is obtained for $H_{c}=8.4$ cm.

The comparison between the two adjustment lengths *d* and $d^{\prime }$, determined with the static and dynamic approaches, shows that its value is the same for both methods. Therefore, in our testing conditions, the soil-structure interaction can be simplified by the proposed cantilever model.

## 7. Conclusions

The present study focused on the effect of scour on the static and dynamic responses of a rod partially embedded in sand. Distributed strain measurement using OFDR technique provided a detailed strain profile along the rod. A theoretical formulation was developed using the Winkler model of the soil and compared to the measured experimental values. The errors did not exceed 7% highlighting the accuracy of the OFDR. The static tests results also showed that the fiber optic sensor performed identically under tension and compression which is crucial when the rod will be deformed by the flow.

In order to monitor scour, the turning point of the strain profile was used to identify the ground level. The theoretical model provides insight of the parameters influencing the maximum stain position along the rod, which are: the lateral stiffness of the soil, the Young modulus and the inertia of the tested rod.

The results also showed that the effect of scour on the strain level is only noticeable near the ground. As scour increases, the value of the strain increases along the first layer of the soil. However, no significant variation was detected for greater depth.

Regarding the dynamic tests, the results showed that the first frequency of the rod decreases significantly with scour depth. Finally, an equivalent cantilever model was proposed for both static and dynamic tests. This model correlates both the natural frequency and the deflection of the rod to its exposed length.

## Figures and Tables

**Figure 1 sensors-20-00321-f001:**
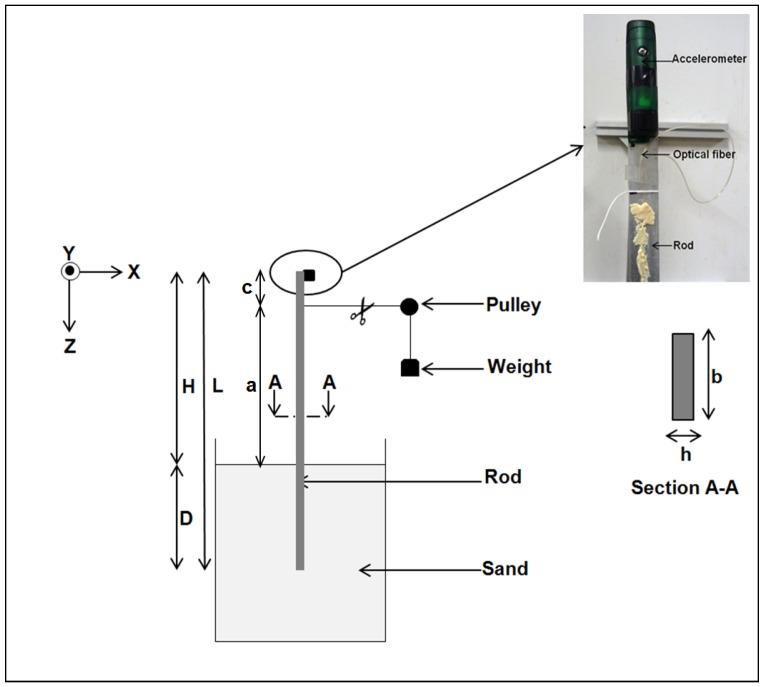
Experimental setup of the static and dynamic tests and close up look to fiber optic installation. *L* is the length of the beam, with *D* the embedded length and *H* the exposed length. *a* is the exccentricity of the load to the soil level. The scissors show where the thread has been cut to apply the lateral force.

**Figure 2 sensors-20-00321-f002:**
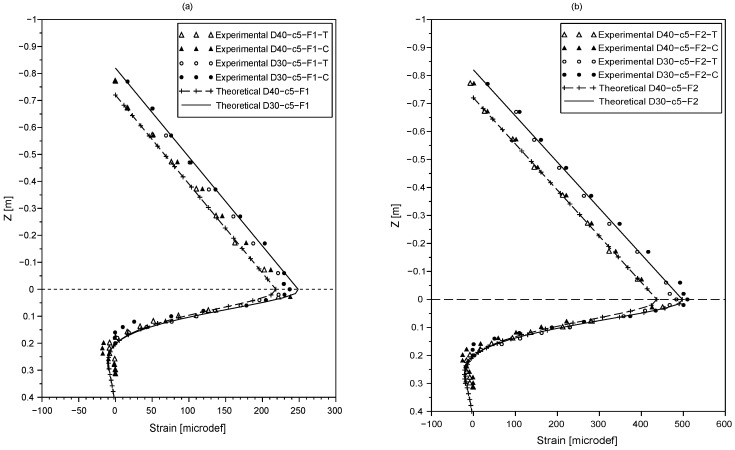
Strain profiles along the rod under tension ‘T’ and compression ‘C’ provided by the fiber optic for the load *F* [(**a**) F=F1 (**b**) F=F2] applied at c=5 cm from the tip of the rod for an embedded length of D=40 cm and D=30 cm.

**Figure 3 sensors-20-00321-f003:**
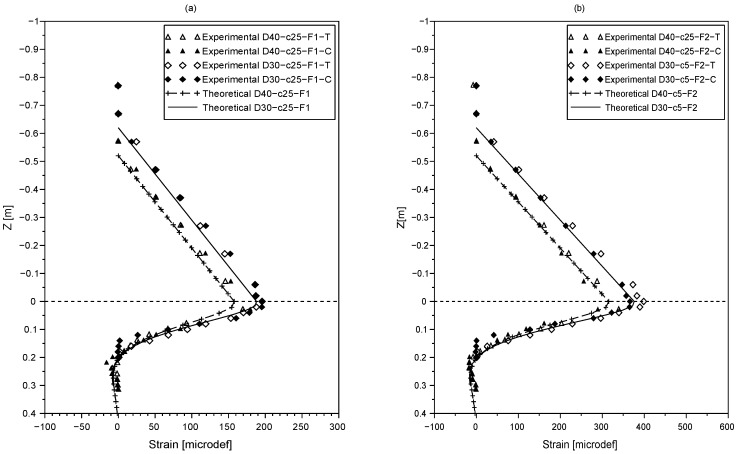
Strain profiles along the rod under tension ‘T’ and compression ‘C’ provided by the fiber optic for the load *F* [(**a**) F=F1 (**b**) F=F2] applied at c=25 cm from the tip of the rod for an embedded length of D=40 cm and D=30 cm.

**Figure 4 sensors-20-00321-f004:**
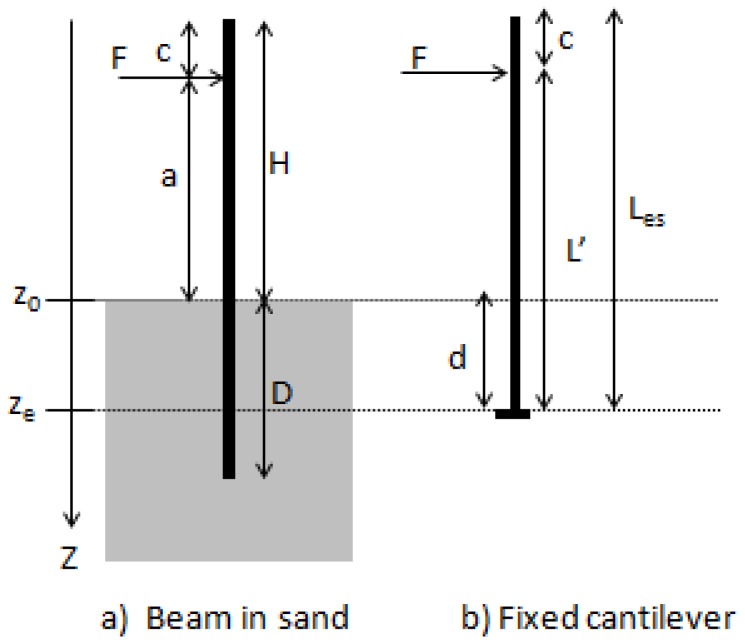
Definition sketch of the static equivalent length. The subfigure (**a**) shows the physical problem, whose static behavior is equivalent to the one of the cantilevered beam of the subfigure (**b**). *L* is the length of the beam, with *D* the embedded length and *H* the exposed length. *a* is the excentricity of the load to the soil level. $z_{0}$ is the soil level, and $z_{e}$ the level of the fixed of the cantilevered, equivalent beam.

**Figure 5 sensors-20-00321-f005:**
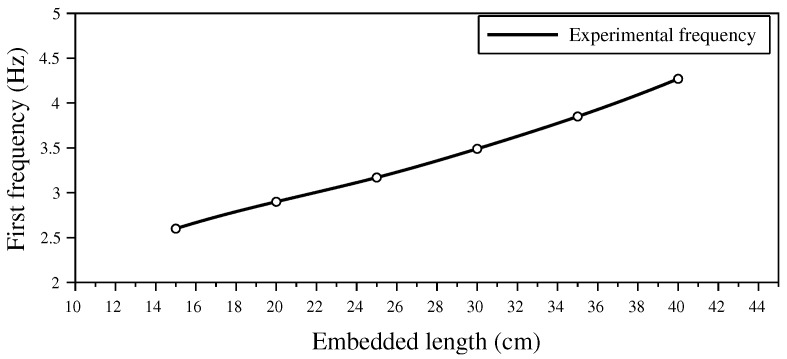
Variation of the first frequency with the embedded length *D* of the rod.

**Figure 6 sensors-20-00321-f006:**
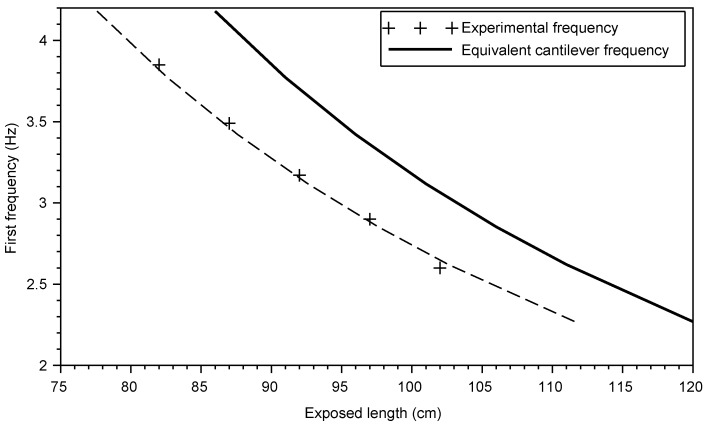
Variation of the first frequency with the exposed length *H* of the rod.

**Table 1 sensors-20-00321-t001:** List of static tests.

Test Name	Embedment Length *D* (cm)	Load Distance from the Tip of the Rod *c* (cm)	Load
$D40-c5-F_{1}$	40	5	$F_{1}$
$D40-c5-F_{2}$	40	5	$F_{2}$
$D30-c5-F_{1}$	30	5	$F_{1}$
$D30-c5-F_{2}$	30	5	$F_{2}$
$D40-c25-F_{1}$	40	25	$F_{1}$
$D40-c25-F_{2}$	40	25	$F_{2}$
$D30-c25-F_{1}$	30	25	$F_{1}$
$D30-c25-F_{2}$	30	25	$F_{2}$

**Table 2 sensors-20-00321-t002:** Theoretical position of the maximum strain.

Test Configuration	$z_{\max}$ (cm)
D40−c5	0.46
D30−c5	0.40
D40−c25	0.62
D30−c25	0.53

**Table 3 sensors-20-00321-t003:** Position Z of the base of equivalent cantilever.

Test Configuration	Deflection *w* at z=−a (cm)	$L^{\prime }$ (cm)	*d* (cm)
$D40-c5-(F_{1}\&F_{2})$	4.2	80.4	8.4
$D30-c5-(F_{1}\&F_{2})$	6.0	90.4	8.4
$D40-c25-(F_{1}\&F_{2})$	1.8	60.4	8.4
$D30-c25-(F_{1}\&F_{2})$	2.8	70.4	8.4

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
