# Peer review of "Distributed Optical Fiber-Based Approach for Soil–Structure Interaction"

_sensors, 2020, doi:10.3390/s20010321_

Round 1
Reviewer 1 Report
1.To solve the differential equation (4), boundary condition M(-a)=0 was employed. Why is the moment=0 at z=-a? It is not free end.
What is the effect of c on the strain distribution along the rod as shown in Figures 2 and 3? Why is the strain in the exposed part of the rod increasing with the decrease of the embedded length D as shown in Figures 2 and 3? It is useful to include the deflection of the rod in the manuscript. In lines 146-147 “as scour increases, the bending moment and strain values increase as a consequence of a greater eccentricity of the applied force F”, can the authors explain this statement in more detail? The "adjustment static length" is the same for all the considered configurations as shown in Table 3. Is any explanation for this conclusion?Author Response
Dear Reviewer,
Thank you for your comments.
I answer here to these comments, and I made corresponding changes in the manuscript (generally by adding the required information).
To solve the differential equation (4), boundary condition M(-a)=0 was employed. Why is the moment=0 at z=-a? It is not free end.
R: Indeed, there is no free end at z=-a. Nevertheless, the applied force has been designed to be a pure force (no punctual couple, so no rotation). Moreover, the displacements are assumend to be small. Therefore, by using the equation linking the second derivative of the displacement and the moment, we can assume the moment to be null.
The words "By assuming that the displacements are small and that the applied force is strictly perpendicaular to the beam" have been added.
What is the effect of c on the strain distribution along the rod as shown in Figures 2 and 3? Why is the strain in the exposed part of the rod increasing with the decrease of the embedded length D as shown in Figures 2 and 3? It is useful to include the deflection of the rod in the manuscript.
R: As c is the parameter giving the point where load F is applied, it is linked to the max displacement and its location (see Equation 12, by using the fact that the exposed length of the beam is equal to a+c).
The strain in the exposed part of the rod is increasing with the decrease of the embedded length D because of the strain (of the beam) continuity at the soil level. Moreover, with lower embedded length, the strain in the embedded part is higher to achieve the equilibrium (Winkler springs on a lower length).
The expression of the deflection of the rod has been added.
In lines 146-147 “as scour increases, the bending moment and strain values increase as a consequence of a greater eccentricity of the applied force F”, can the authors explain this statement in more detail?
R: (for lines 146-147) as scour increases, the bending moment and strain values increase as a consequence of a greater eccentricity of the applied force F, as the exposed length of the beam increases (scour => less embedded length => higher exposed length).
The corresponding text has been added to the manuscript.
The "adjustment static length" is the same for all the considered configurations as shown in Table 3. Is any explanation for this conclusion?
R: Indeed, it means that for a given beam and a given soil, a certain embedded length ensures that the beam is completely fixed.
This is a first result.
The other one, maybe even more interesting, is that this equivalent length is valid for both the static and the dynamic behavior of the beam. The work here is dealing with the static equivalent length. An article has already been published by the authors on the dynamic equivalent length. Work is still ongoing to derive an analytical expression for this equivalent length.
A sentence has been added in this conclusion to highlight this ("This result means that for a given range a embedded length, the partially embedded beam can be considered statically as a cantilevered beam of a higher length, to take into account the embedded length that is needed to support the beam. " line 190, page 9).
Hoping that this explanations and changes are satisfying for remarks,
Best regards
Reviewer 2 Report
The authors present an interesting work regarding the application of fiber-based sensors to obtain information regarding scour of submersed structures. They use Frequency Domain Reflectometry and strain measurements to accomplish the goal.
In the overall, the paper is well written. Language is clear and images correctly complement the text. Having an analytical model to obtain data and compare with experimental implementation of the sensors gives a good scientific soundness and validates the methodology.
However, the paper’s organization requires some improvement. In particular. The “theoretical formulation” (section 3.1) should not appear as a “result”. The theory behind modelling should appear in a specific section, after introduction and before “materials and methods”. The section regarding the results should present only… the results. For all experimental and theoretical procedures. In the analysis section, everything is analysed, including a comparison between theoretical and experimental data.
Besides, the description of the setup for the application OFDR technique should appear in the section “Fiber optic sensing technology”. A schematic (and corresponding reference in the text) should include the overall sensing system and proceed the schematic of Fig. 1. The latter only addresses the final part of the setup, while the interrogation is forgot. The new schematic should clearly indicate the positions of the different devices mentioned in the text (e.g. the OBR).
Also, in figure 1, what is the meaning of the scissor? And, related with this figure, although the letters that appear are explained in the “Abbreviations” section, the legend should direct the reader for this section. This is even more important since the authors do not mention most of these symbols in the corresponding explaining text. The same comment applies for figure 4.
A final remark regarding the “Winkler approach” presented in the beginning of the theoretical formulation. The reference is too old (1867!), and should be complemented with (at least) a more recent one addressing the technique.
Author Response
Dear Reviewer,
Thank you for your comments.
I answer here to these comments, and I made corresponding changes in the manuscript (generally by adding the required information).
In the overall, the paper is well written. Language is clear and images correctly complement the text. Having an analytical model to obtain data and compare with experimental implementation of the sensors gives a good scientific soundness and validates the methodology.R: The language and the typo have been checked.
However, the paper’s organization requires some improvement. In particular. The “theoretical formulation” (section 3.1) should not appear as a “result”. The theory behind modelling should appear in a specific section, after introduction and before “materials and methods”. The section regarding the results should present only… the results. For all experimental and theoretical procedures. In the analysis section, everything is analysed, including a comparison between theoretical and experimental data.
Besides, the description of the setup for the application OFDR technique should appear in the section “Fiber optic sensing technology”. A schematic (and corresponding reference in the text) should include the overall sensing system and proceed the schematic of Fig. 1. The latter only addresses the final part of the setup, while the interrogation is forgot. The new schematic should clearly indicate the positions of the different devices mentioned in the text (e.g. the OBR).
R: You are right.
The manuscript has been modified to comply with these precise demands.
Also, in figure 1, what is the meaning of the scissor? And, related with this figure, although the letters that appear are explained in the “Abbreviations” section, the legend should direct the reader for this section. This is even more important since the authors do not mention most of these symbols in the corresponding explaining text. The same comment applies for figure 4.
R: The scissor are showing where the threated has been cut (with scissors) to apply the lateral force. Text to explain this has been added.
Moreover, the letters have been explained in the legend, as you requested.
A final remark regarding the “Winkler approach” presented in the beginning of the theoretical formulation. The reference is too old (1867!), and should be complemented with (at least) a more recent one addressing the technique.
R: True! We cited an old text to show that this theory is not new, at all! But you are right, we added another, newer reference to show that this theory is still used, in research, academic and engineering works.
Hoping that this explanations and changes are satisfying for remarks,
Best regards
Round 2
Reviewer 1 Report
I am satisfied with the revisions made by the authors.